# Sentience and the Origins of Consciousness: From Cartesian Duality to Markovian Monism

**DOI:** 10.3390/e22050516

**Published:** 2020-04-30

**Authors:** Karl J. Friston, Wanja Wiese, J. Allan Hobson

**Affiliations:** 1The Wellcome Centre for Human Neuroimaging, Institute of Neurology, Queen Square, London WC1N 3AR, UK; 2Department of Philosophy, Johannes Gutenberg University Mainz, Jakob-Welder-Weg 18, 55128 Mainz, Germany; 3Division of Sleep Medicine, Harvard Medical School, 74 Fenwood Road, Boston, MA 02115, USA

**Keywords:** consciousness, information geometry, Markovian monism

## Abstract

This essay addresses Cartesian duality and how its implicit dialectic might be repaired using physics and information theory. Our agenda is to describe a key distinction in the physical sciences that may provide a foundation for the distinction between mind and matter, and between sentient and intentional systems. From this perspective, it becomes tenable to talk about the physics of sentience and ‘forces’ that underwrite our beliefs (in the sense of probability distributions represented by our internal states), which may ground our mental states and consciousness. We will refer to this view as Markovian monism, which entails two claims: (1) fundamentally, there is only one type of thing and only one type of irreducible property (hence *monism*). (2) All systems possessing a Markov blanket have properties that are relevant for understanding the mind and consciousness: if such systems have mental properties, then they have them partly by virtue of possessing a Markov blanket (hence *Markovian*). Markovian monism rests upon the information geometry of random dynamic systems. In brief, the information geometry induced in any system—whose internal states can be distinguished from external states—must acquire a dual aspect. This dual aspect concerns the (intrinsic) information geometry of the probabilistic evolution of internal states and a separate (extrinsic) information geometry of probabilistic beliefs about external states that are parameterised by internal states. We call these intrinsic (i.e., mechanical, or state-based) and extrinsic (i.e., Markovian, or belief-based) information geometries, respectively. Although these mathematical notions may sound complicated, they are fairly straightforward to handle, and may offer a means through which to frame the origins of consciousness.

## 1. Introduction

The aim of this essay is to emphasise a couple of key technical distinctions that seem especially prescient for an understanding of the beliefs and intentions that underpin pre-theoretical notions of consciousness. What follows is an attempt to describe constructs from information theory and physics that place certain constraints on the dynamics of self-organising creatures, such as ourselves. These constraints lend themselves to an easy interpretation in terms of beliefs and intentions; provided one defines their meaning carefully in relation to the mathematical objects at hand. The benefit of articulating a calculus of beliefs (and intentions) from first principles has yet to be demonstrated; however, just having a calculus of this sort may provide useful perspectives on current philosophical debates. Furthermore, trying to articulate pre-theoretical notions in terms of maths should, in principle, expand the scope of dialogue in this area. To illustrate this, we will try to license talk about physical forces causing beliefs in a non-mysterious way—a way that clearly identifies systems or artefacts that are and are not equipped with processes that can ground mental capacities and consciousness.

To make a coherent argument along these lines, it will be necessary to introduce a few technical concepts. The formal basis of the arguments in this—more philosophical—treatment of sentience and physics can be found in [1]. The current paper starts were Friston (ibid.) stops; namely, to examine the philosophical implications of Markov blankets and the ensuing Bayesian mechanics. For readers who are more technically minded, the derivations and explanations of the equations in this paper can be found in [1] (using the same notation). We have attempted to unpack the derivations for non-mathematical readers but will retain key technical terms, so that the lineage of what follows can be read clearly. To avoid cluttering the narrative with definitions, a glossary of terms and expressions is provided at the end of the paper. In brief, we first establish the basic setup used to describe physical systems that evince the phenomenology necessary to accommodate pre-theoretical notions of consciousness. This will involve the introduction of Markov blankets and the distinction between the internal and external states of a system or creature.

Having established the distinction between external and internal states, we introduce the notion of information length and information geometry. This is the first key move in the theoretical analysis on offer. Crucially, information geometry allows us to establish a calculus of beliefs in terms of probability distributions. This calculus enables a distinction to be made between the probability distribution *about things* and the probability distribution *of things*. This distinction is then treated as one way of describing an account that (literally) maps belief states onto physical states; here, beliefs about external states that are parameterised, represented, encoded or coherent with internal states. We shall call the ensuing view *Markovian monism* because it is predicated on the existence of a Markov blanket. 

This brings us to a modest representationalism[note 1], which allows one to talk about flows, energy gradients and forces that shape the dynamics of internal states and, necessarily, the beliefs they parameterise. The next section considers the nature of these beliefs and, in particular, beliefs about how internal states couple to external states; namely, beliefs about action upon the world ‘out there’. To do this formally, we have to look at two distinct ways of describing the dynamics and introduce the notion of trajectories via the path integral formulation. Having done this, we can then associate intentions with beliefs about action—that, in turn, depend upon beliefs about the consequences of action. At this point, we can make a distinction between systems that have a rudimentary information geometry of a reflexive, instantaneous sort—and systems that hold beliefs about the future. It is this quantitative distinction that may provide a spectrum of intentional or agential systems, ranging from protozoa to people. We conclude with a brief discussion of related formulations—and how the central role of sentience, observation, measurement, or inference opens the door for further developments of a sentient physics. In particular, we will discuss how Markovian monism can be interpreted in terms of existing theories regarding the relationship between mind and matter, such as neutral monism and panprotopsychism.

The primary target of this paper is sentience. Our use of the word “sentience” here is in the sense of “responsive to sensory impressions”. It is not used in the philosophy of mind sense; namely, the capacity to perceive or experience subjectively, i.e., phenomenal consciousness, or having ‘qualia’. Sentience here, simply implies the existence of a non-empty subset of systemic states; namely, sensory states. In virtue of the conditional dependencies that define this subset (i.e., the Markov blanket partition), the internal states are necessarily ‘responsive to’ sensory states and thus the dictionary definition is fulfilled. The deeper philosophical issue of sentience speaks to the hard problem of tying down quantitative experience or subjective experience within the information geometry afforded by the Markov blanket construction. We will return to this below.

While most of this paper deals with sentience in the sense just specified, it may shed light on the origins of consciousness. First, applying the concept of subjective, phenomenal consciousness to a system trivially presupposes that this system can be described from two perspectives (i.e., from a third- and from a first-person perspective). Second, the minimal form of goal-directedness and ‘as if’ intentionality—that one can ascribe to sentient systems—provide conceptual building blocks that ground more high-level concepts, such as physical computation, intentionality, and representation, which may be useful to understand the evolutionary transition from non-conscious to conscious organisms, and thereby illuminate the origins of consciousness.

## 2. Markov Blankets and Self-Organisation

Before we can talk about anything, we have to consider what distinguishes a ‘thing’ from everything else. Mathematically, this requires the existence of a particular partition of all states a system could be in into external, (Markov) blanket and internal states. A Markov blanket comprises a set of states that renders states internal to the blanket conditionally independent of external states. The term was originally coined by Pearl in the context of Bayesian networks [2]. For a Bayesian network (i.e., a directed acyclic graphical model) the Markov blanket comprises the parents, children, and parents of the children of a state or node. For a Markov random field (i.e., an undirected graphical model), the Markov blanket comprises the parents and children, i.e., its neighbours. For a dependency network (i.e., a directed cyclic graphical model) the Markov blanket comprises just the parents. For treatments of Markov blankets in the life sciences, please see [3,4,5,6,7,8]. The three-way partition induced by the Markov blanket enables one to distinguish internal and external states via their conditional independence, given blanket states. The blanket states themselves can be further partitioned into sensory and active states, where sensory states are not influenced by internal states and active states are not influenced by external states [9]. Note that all we have done here is to stipulatively define a ‘thing’ in terms of its internal states (and Markov blanket) in terms of what does *not* influence what. The requisite absence of specific influences are precisely those described above; namely, internal states and external states only influence each other via the Markov blanket, while sensory states are not influenced by internal states, a similar relationship is true for active and external states. A key insight here is that structure emerges from influences that *are not there*, much like a sculpture emerges from the material removed. There are lots of interesting implications of defining things in terms of Markov blankets (please see Figure 1 for a couple of intuitive examples); however, we will place the notion of a Markov blanket to one side for the moment and consider how systemic states behave in general. After this, we will then consider the implications of this generic behaviour, when there is a Markov blanket play.

## 3. The Langevin Formalism and Density Dynamics

Starting from first principles, if we assume that a system exists, in the sense that it has measurable characteristics over some nontrivial period of time,[note 2] then we can express its evolution in terms of a random dynamical system. This just means that the system can be described in terms of changes in states over time that are subject to some random fluctuations:(1)$$\dot{x}(\tau )=f(x,\tau )+\omega .$$

This is a completely general specification of (Langevin) dynamics that underwrites nearly all of physics [10,11,12]. In brief, the dynamics in (1) can be described in terms of two equivalent formulations—the dynamics of the accompanying probability density over the states and the path integral formulation.[note 3]

We will be interested in systems that have measurable characteristics, which means that they must converge to some attracting set or manifold, known as a random or pullback attractor [13].[note 4] After a sufficient period of time, as the system evolves, it will trace out a trajectory—in state space—that circulates, usually in a highly itinerant fashion, on the attracting manifold. This means that if we observe the system at random, there is a certain probability of finding it in a particular state. This is known as the nonequilibrium steady-state density [12].

It is natural to ask whether a single attracting manifold is an appropriate construct to describe a system or creature over its lifetime; especially when certain ‘life-cycles’ have distinct developmental stages or indeed feature metamorphosis. From the perspective of the current argument, it helps to appreciate that the attracting manifold is itself a random set.[note 5] In other words, a particle or person is never ‘off’ their manifold—they just occupy states that are more or less likely, given the kind of thing they are (i.e., something’s *characteristic* states are an attracting set of states that it is likely to occupy). Technically, this peripatetic itinerancy corresponds to stochastic chaos, where excursions from the attracting set—driven by random fluctuations—are an integral aspect of the dynamics. These excursions are repaired through the flow that counters the effects of random fluctuations and underwrites the information geometry of self-organisation. This formulation can, in principle, accommodate slow changes to the attracting set—and implicit Markov blanket—that may require the notion of wandering sets [14].

The reason that this is interesting is that one can use standard descriptions of density dynamics to express the flow of states as a gradient flow on something called *self-information* or *surprisal* [15,16,17,18]. Without going into details, this is the steady-state solution to the Fokker Planck equation [19,20,21,22,23]. This equation says that, on average, the states of any system with an attracting set must conform to a gradient flow on surprisal; namely, the negative logarithm of the probability density at nonequilibrium steady state [24,25].
(2)$$\begin{array}{l} f(x)=(Q-\Gamma )\cdot \nabla I(x)\\ I(x)=-\ln p(x) \end{array}$$

This is the solution to the Fokker-Planck equation when the system has attained nonequilibrium steady-state. It says that the average flow of systemic states has two parts. The first (gradient) component involves surprisal gradients, while the second circulates on iso-probability contours. The gradient flow effectively counters the dispersion due to random fluctuations, such that the probability density does not change over time. See Figure 2 for an intuitive illustration of this solution.

The key move now is to put the Markov blanket back in play. The above equation holds (nontrivially) for the internal, blanket, and external states, where we can drop the appropriate states from the gradient flows, according to the specification of the Markov blanket in Figure 1. In particular, if we just focus on internal and active states—which we will refer to as *autonomous* states—we have the following flows[note 6] (see p. 17 and pp. 20,21 in [1]).
(3)$$\begin{array}{c} f_{\alpha }(\pi )=(Q_{\alpha \alpha }-\Gamma_{\alpha \alpha })\nabla_{\alpha }I(\pi )\\ \alpha =\{a,\mu \}\\ \pi =\{s,\alpha \} \end{array}$$

This means anything that can be measured (i.e., a system with a Markov blanket and attracting set) must possess the above gradient flows. In turn, this means that internal and active states will look as if they are trying to minimise exactly the same quantity; namely, the surprisal of states that constitute the thing, particle, or creature. These are the internal states and their Markov blanket; i.e., *particular states*.[note 7] This means that anything that exists must, in some sense, be self-evidencing [37].

**Figure 3 entropy-22-00516-f003:**
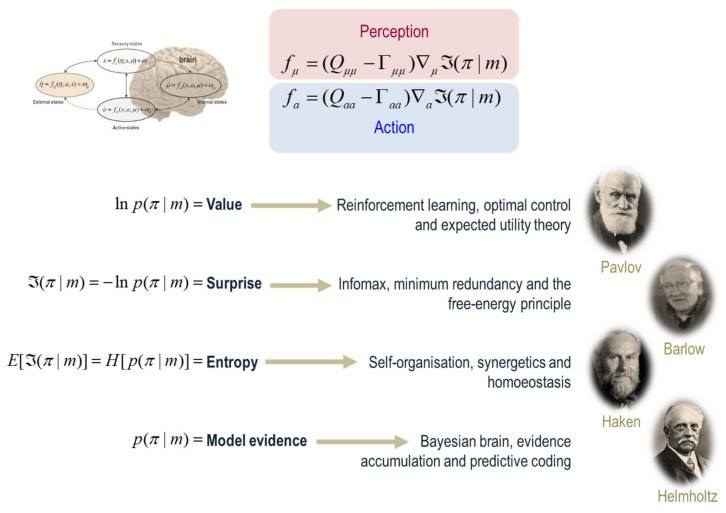
(*Markov blankets and other formulations*): This schematic illustrates the various interpretations of a gradient flow on surprisal. Recall that the existence of a Markov blanket implies a certain lack of influences among internal, blanket, and external states. At nonequilibrium steady-state, these independencies have an important consequence; internal and active states are the only states that are not influenced by external states, which means their dynamics (i.e., perception and action) are a function of, and only of, particular states; i.e., a particular surprisal.[note 8] This surprisal has a number of interesting interpretations. Given it is the negative log probability of finding a particle or creature in a particular state, minimising particular surprisal corresponds to maximising the *value* of a particle’s state. This interpretation is licensed by the fact that the states with a high probability are, by definition, attracting states. On this view, one can then spin-off an interpretation in terms of reinforcement learning [30], optimal control theory [31] and, in economics, expected utility theory [39]. Indeed, any scheme predicated on the optimisation of some objective function can now be cast in terms of minimising a particular surprisal—in terms of perception and action (i.e., the flow of internal and active states). The minimisation of particular surprisal leads to a series of influential accounts of neuronal dynamics; including the principle of maximum mutual information [40,41], the principles of minimum redundancy and maximum efficiency [33] and—as we will see later—the free energy principle [42]. Crucially, the average or expected surprisal (over time or particular states of being) corresponds to entropy. This means that action and perception look as if they are minimising a particular entropy. The implicit resistance to the second law of thermodynamics leads us to theories of self-organisation, such as synergetics in physics [29,43,44] or homoeostasis in physiology [35,45,46]. Finally, the probability of any particular states given a Markov blanket (*m*) is, on a statistical view, model evidence [18,47]. This means that all the above formulations are internally consistent with things like the Bayesian brain hypothesis, evidence accumulation and predictive coding; most of which inherit from Helmholtz’s motion of unconscious inference [48], later unpacked in terms of perception as hypothesis testing in 20th century psychology [49] and machine learning [50]. In short, the very existence of something leads in the natural way to a whole series of optimisation frameworks in the physical and life sciences that lends each a construct validity in relation to the others.

## 4. Bayesian Mechanics

Thus, if we can describe anything as self-evidencing—in the sense of possessing a dynamics that tries to minimise a particular surprisal—or maximise a particular model evidence, what is the model? It is at this point we get into the realm of inference and Bayesian mechanics, which follows naturally from the density dynamics of the preceding section. The key move here rests upon another fundamental but simple consequence of possessing a Markov blanket.

Technically, the stipulative existence of a Markov blanket means that internal and external states are conditionally independent of each other, when conditioned on blanket states. This has an important consequence. In brief, for every given blanket state there must exist a density over internal states and a density over external states. The former must possess an expectation (i.e., average) or mode (i.e., maximum). This means for every conditional expectation of internal states there must be a conditional density over external states. In short, the mapping between the expected (i.e., average) internal state (for any given blanket state) and a conditional density over external states (i.e., a Bayesian belief about external states) inherits from the conditional independencies that define a Markov blanket. In turn, anything that exists is defined by its Markov blanket. A more formal treatment of this can be found on p. 84 of [1]. See also [3,38] for further discussion.

Therefore, if internal and external states are conditionally independent, then for every given blanket state there is an expected internal state and a conditional probability density over external states. In other words, there must be a one-to-one relationship between the average internal state of a particle (or creature) and a probability density over external states, for every given blanket state.[note 9] This means that we can express the posterior or conditional density over external states as a probabilistic belief that is parameterized by internal states:(4)$$\begin{array}{c} q_{\mu }(\eta )=p(\eta |b)=p(\eta |\pi )\\ \mu (b)≜\arg\max_{\mu }p(\mu |b) \end{array}$$

On the assumption that the number or dimensionality of internal states is greater than the number of blanket states, the dimensionality of the internal (statistical) manifold—defined by the second equality in (4)—corresponds to the dimensionality of blanket states (which ensures an injective and surjective mapping). This is important because it means there is a subspace (i.e., statistical manifold) of internal states whose dimensionality corresponds to dimensionality of the blanket state (e.g., cardinality of sensory receptors). Heuristically, this means that many external states of affairs can only be represented probabilistically; in a way that depends upon the number of blanket states. Furthermore, the states parameterising this conditional density are conditional expectations; namely, the average internal state, for each blanket state—please see Figure 18 in [1] for a worked (numerical) example.

This is important from a number of perspectives. First, it allows us to interpret the flow of (expected) autonomous states (i.e., action and perception) as a gradient flow on something called variational free energy.[note 10]
(5)$$\begin{array}{cc} f_{\alpha }(\pi ) & \approx (Q_{\alpha \alpha }-\Gamma_{\alpha \alpha })\nabla_{\alpha }F(\pi )\\ F(\pi ) & \geq I(\pi )\\ F(\pi ) & ≜\underset{energy}{\underset{︸}{E_{q}[I(\eta ,\pi )]}}-\underset{entropy}{\underset{︸}{H[q_{\mu }(\eta )]}}\\ & =\underset{surprisal}{\underset{︸}{I(\pi )}}+\underset{bound}{\underset{︸}{D[q_{\mu }(\eta )||p(\eta |\pi )]}}\\ & =\underset{inaccuracy}{\underset{︸}{E_{q}[I(\pi |\eta )]}}+\underset{complexity}{\underset{︸}{D[q_{\mu }(\eta )||p(\eta )]}} \end{array}$$

The second thing that (4) brings to the table is an *information geometry* and attending calculus of beliefs. From now on, we will associate beliefs with the probability density above that is parameterised by (expected) internal states. Note that these beliefs are non-propositional, where ‘belief’ is used in the sense of ‘belief propagation’ and ‘Bayesian belief updating’ that can always be formulated as minimising variational free energy [51,52,58]. To license a description of this conditional density in terms of beliefs, we can now appeal to information geometry [23,59,60,61].

## 5. Information Geometry and Beliefs

Information geometry is a formalism that considers the metric or geometrical properties of statistical manifolds. Generally speaking, a collection of points in some arbitrary state space does not, in and of itself, have any geometry or associated notion of distance, e.g., one cannot say whether one point is near another. To equip a space with a geometry, one has to supply something called a metric tensor–such that small displacements in state space can be associated with a metric of distance. For familiar Euclidean spaces, this metric tensor is the identity matrix. In other words, moving one centimetre in this direction means that I have moved a distance of 1 cm. However, generally speaking, metric spaces do not have such a simple tensor form[note 11]. Provided the metric tensor is symmetrical and positive (for all dimensions of the states in question), the geometry is said to be Riemannian. So, what is special about the Riemannian geometry of statistical manifolds?

A statistical manifold is a special state space, in which the states represent the parameters of a probability distribution. For example, a two-dimensional manifold, whose coordinates are mean and precision, would constitute a statistical manifold for Gaussian distributions. In other words, for every point on the statistical manifold there would be a corresponding Gaussian (bell shaped) probability density. The important thing here is that any statistical manifold is necessarily equipped with a unique metric tensor, known as the Fisher information metric [23,59,62].[note 12]
(6)$$\begin{array}{c} d\ell^{2}=g_{ij}d\mu^{i}d\mu^{j}\\ g(\mu )=\nabla_{\mu^{\prime }\mu^{\prime }}D[q_{\mu^{\prime }}(\eta )||q_{\mu }(\eta )]{|}_{\mu^{\prime }=\mu }=E_{q}[\nabla_{\mu }\ln q_{\mu }(\eta )\times \nabla_{\mu }\ln q_{\mu }(\eta )] \end{array}$$

Here, dℓ is the information length associated with small displacements on the statistical manifold $d\mu =\mu^{\prime }-\mu$ induced by a probability density $q_{\mu }(\eta )$. It is not important to understand the details of this metric; other than to note that it must exist. In brief, the distance between two points on the statistical manifold obtains by accumulating the Kullback-Leibler divergence between the probability distributions encoded as we move along a path from one point to another. In other words, the information length scores the number of different probabilistic or belief states encountered in moving from one part of a statistical manifold to another. The path with the smallest length is known as a geodesic. So why is this interesting?

If we return to the independencies induced by the Markov blanket, Equation (4) tells us something fundamental. The (expected) internal states have acquired an information geometry, because they parameterise probabilistic beliefs about external states. This geometry is uniquely supplied by the Fisher information metric specified by the associated beliefs. In short, we now know that there is a unique geometry in some belief space that can be associated with the internal (physical) state of any particle or creature. Furthermore, we also know that the gradient flows describing the dynamics of internal states can be expressed as a gradient flow on a variational free energy functional (i.e., function of the function) *of beliefs*: see (5). All this follows from first principles and yet we have something quite remarkable in hand: if anything exists, its autonomous states will (appear to) be driven by gradient forces established by an information geometry or, more simply, probabilistic beliefs.[note 13] From (5):(7)$$\begin{array}{c} f_{\alpha }(\pi )\approx (Q_{\alpha \alpha }-\Gamma_{\alpha \alpha })\nabla_{\alpha }F\\ F(\pi )\equiv F[s,q_{\mu }(\eta )] \end{array}$$

We will call the information geometry that follows from this an *extrinsic* information geometry because it rests upon probabilistic (Bayesian) beliefs about external states. Bayesian beliefs are just conditional probability distributions *that are manifest* in the sense of being encoded by the (internal) states of a physical system. This means it would be perfectly sensible to say that a bacterium has certain Bayesian beliefs about the extracellular milieu—that are encoded by intracellular states. Similarly, in a brain, neuronal activity in the visual cortex parameterizes a Bayesian belief about some visible attribute of the sensorium. Clearly, these kinds of beliefs are not propositional in nature.

Things get even more interesting when we step back and think about the density dynamics of the internal states. Recall from above, that an information geometry is a necessary property of any statistical manifold constituted by parametric states. So, are there any parameters of the probability density over the internal states themselves? The answer here is yes. In fact, these parameters are thermodynamic variables (e.g., pressure) that underwrite thermodynamics or statistical mechanics [62,64]. An important parameter of this kind is time itself. This follows because if we start the internal states from any initial probability density, it will evolve over time to its non-equilibrium steady-state solution. Crucially, this means that we can parameterise the density over internal states with time—and time becomes our statistical manifold. This leads to the challenging intuition, that distance travelled in time can change as we move into the future. In virtue of the existence of the attracting set, the increase in this information length will eventually slow down and stop (as the probability density in the distant future approaches its nonequilibrium steady state)[note 14]. In turn, the information length furnishes a useful measure of distance from any initial conditions to nonequilibrium steady-state–that has been exploited in characterising self-organisation in random, chaotic dynamical systems [23,62]. We will refer to the accompanying information geometry as an *intrinsic* geometry, because it is intrinsic to the density dynamics of the states *per se*.[note 15] From our point of view, this means there are two information geometries in play with the following metrics: (8)$$\begin{array}{cc} g(\tau )=\nabla_{\tau^{\prime }\tau^{\prime }}D[p_{\tau^{\prime }}(\mu )||p_{\tau }(\mu )]{|}_{\tau^{\prime }=\tau } & \mathrm{intrinsic}\\ g(\mu )=\nabla_{\mu^{\prime }\mu^{\prime }}D[q_{\mu^{\prime }}(\eta )||q_{\mu }(\eta )]{|}_{\mu^{\prime }=\mu } & \mathrm{extrinsic} \end{array}$$

First, there is an intrinsic information geometry inherent in the information length based upon time-dependent probability densities over internal states. This information length characterises the system or creature in terms of itinerant, self-organising density dynamics that forms the basis of statistical mechanics in physics, i.e., a physical, material, or *mechanical* information geometry that is *intrinsic* to the system. At the same time, there is an information geometry in the space of internal states that refers to belief distributions over external states. This is the *extrinsic* information geometry that inherits from the *Markovian* conditions that define, stipulatively, autonomous states (via their Markov blanket). The extrinsic geometry is conjugate to the intrinsic geometry but measures distances between beliefs. Both are measurable, and both supervene on the same Langevin dynamics. 

Again, this is not mysterious it is just a mathematical statement of the way things are. What is interesting here is that internal states have a dual aspect information geometry that seems to be related to the dual aspect monism—usually advanced to counter Cartesian (matter and mind) duality. On a simple interpretation, one might associate the information length of internal states with the material behaviour of particles or creatures, while the mindful aspects are naturally associated with the probabilistic beliefs that underwrite the extrinsic information geometry of internal states. However, the existence of a dual aspect information geometry does not, in and of itself, give a system mental states and consciousness, but only computational properties (including probabilistic beliefs). Furthermore, the extrinsic information geometry is ultimately reducible to the intrinsic information geometry (and the other way around), in the sense that there is a necessary link between them cf. [65], pp. 11–13. Still, physical, and computational properties are not identical.[note 16]

An interesting special case arises if we assume that the conditional beliefs are Gaussian in form (denoted by N in equation (9) below). In this instance, the Fisher information metric becomes the curvature or ‘deepness’ of free energy minima, which is the same as the precision (i.e., inverse covariance) of the beliefs *per se*.
(9)$$\begin{array}{l} q(\mu )=\sum {\left(\mu \right)}^{-1}=\nabla_{\mu \mu }F=-\nabla_{\mu \mu }\ln q(\eta )\\ q(\eta )=N(\sigma (\mu ),\sum (\mu )) \end{array}$$

In other words, distances in belief space depend upon conditional precision or the confidence ascribed to beliefs about external states of affairs ‘out there’. We will return to this interesting case in the conclusion. At the moment, notice that we have a formal way of talking about the ‘force of evidence’ in moving beliefs and how the degree of movement depends upon conditional precision, confidence, or certainty [67,68,69].

## 6. A Force to Be Reckoned with

To make all this concrete, it is perfectly permissible to express the gradient flows in terms of forces supplied by the extrinsic, belief-based information geometry. This just requires a specification of the units of the random fluctuations in terms of Boltzmann’s constant. This means that we can rewrite (7) in terms of a thermodynamic potential U(π) and associated forces $f_{m}(\pi )$, where, at nonequilibrium steady-state (see pp. 65–67 in [1]):(10)$$\begin{array}{cc} f_{\alpha }(\pi ) & =(\mu_{m}-Q_{m})f_{m}(\pi )\\ & =(Q_{m}-\mu_{m})\nabla U(\pi )\\ & \approx (Q_{\alpha \alpha }-\Gamma_{\alpha \alpha })\nabla_{\alpha }F(\pi )\\ f_{m}(\pi ) & ≜-\nabla U(\pi )\\ Q_{\alpha \alpha } & ≜k_{B}T\cdot Q_{m}\\ \Gamma_{\alpha \alpha } & ≜k_{B}T\cdot \mu_{m}\\ U(\pi ) & ≜k_{B}T\cdot I(\pi )\\ & \approx k_{B}T\cdot F(\pi ) \end{array}$$

The last equality is known as the Einstein–Smoluchowski relation, where $\mu_{m}$ is a mobility coefficient. This means, we have factorised the amplitude of random fluctuations $\Gamma_{\alpha \alpha }=\mu_{m}k_{B}T$ into mobility and temperature [12]. Nothing has changed here. All we have done is assign units of measurement to the amplitude of random fluctuations, so that we can interpret the ensuing flow as responding to a force, which can be interpreted as a gradient established by a thermodynamic potential. This thermodynamic potential is just (scaled) surprisal or our free energy functional of beliefs.

These equalities cast the appearance of Cartesian duality in pleasingly transparent terms. The forces that engender our physical dynamics can either be expressed as thermodynamic forces or as self-evidencing; in virtue of the extrinsic information geometry supplied by variational free energy. Mathematically, this duality arises from the fact that the surprisal and variational free energy are conjugate: one rests upon the probability of particular states, while the other is a functional of blanket states and beliefs that are parameterised by internal states. They are conjugate in that they refer to probability densities over conditionally independent (i.e., orthogonal) states; namely, internal and external states.

The point here is that there is no difficulty in moving between descriptions afforded by statistical thermodynamics and self-evidencing (i.e., minimising variational free energy). On this reading, variational free energy is a feature of an extrinsic information geometry induced by beliefs encoded by internal states that have an intrinsic information geometry. This free energy has gradients that exert forces on internal states so that they come to parameterise new beliefs. These new beliefs depend upon blanket (e.g., sensory) states; thereby furnishing a mathematical image of perception. Furthermore, the same Bayesian mechanics applies to active states that change external states—and thereby mediate action upon the world. So, is there anything more to the story?

## 7. Active Inference and the Future

Active inference will become a key aspect of the arguments below, when thinking about different kinds of generative models; specifically, generative models of the consequences of action. On the above arguments, anything (that exists in virtue of possessing a Markov blanket) can be cast as performing some elemental form of inference—and possessing an implicit generative model. However, not all generative models are equal; in the sense that no two things are the same. Later, we will look at special kinds of generative models that underwrite active inference.

Above, we introduced variational free energy as an expression of particular surprisal. This variational form is a functional of sensory states and a conditional density or belief distribution encoded by internal states. However, the variational free energy also depends upon the surprisal of joint particular and external states, I(η,π)≡−lnp(η,π), see (5). On a statistical view, the corresponding nonequilibrium steady-state density p(η,π) is known as a *generative model*. In other words, it constitutes a probabilistic specification of how external and particular states manifest. It is this generative model that licenses an interpretation of particular surprisal in terms of Bayesian mechanics and self-evidencing [37]. So, what does this mean for our formulation of beliefs and intention? 

Note that we can always describe the dynamics of internal states in terms of a gradient flow on variational free energy. This means that the dynamical architecture of any particle or creature can also be expressed as a functional of some generative model that, in some sense, must be isomorphic with the nonequilibrium steady-state density. This has some interesting implications: from the point of view of self-organisation, it tells us immediately that if we interpret the action of a particle or creature in terms of self-evidencing, it says that the implicit generative model—which supplies the forces that change internal and active (i.e., autonomous) states—must be a sufficiently good model of systemic states. This is exactly the good regulator theory that emerged in the formulations of self-organisation at the inception of cybernetics [45,70].[note 17]


## 8. Active Inference and the Path Integral Formulation

We will first preview, heuristically, the final argument in this essay. Because active states depend upon internal states (and the beliefs that they parameterise)—but active states do not depend upon external states—it will look as if particles or creatures are acting on the basis of their beliefs about external states. Furthermore, if a particle or creature acts in a dextrous, precise and adaptive way to fluctuations in its blanket states, it will look as if it is acting to minimise its particular surprisal (or variational free energy). In other words, it will look as if it is trying to minimise the surprisal, expected following an action. This means, it would look as if it is behaving to minimise expected surprisal or self-information, which is uncertainty or its particular entropy. 

Anthropomorphically, a creature will therefore (appear to) have beliefs about the consequences of its action, which means it must have beliefs about the future. So how far into the future? One can formalise a response to this question by turning to the path integral formulation of random dynamical systems [12,13,77,78]. In this formulation, we are not concerned with the probability density over states but rather over trajectories or sequences of states. Specifically, we are interested in the probability of trajectories of autonomous states, often referred to as ‘policies’ in the optimal control literature [32]. So, what can one say about the probability of different courses of action in the future? 

We can now turn to the information length associated with the evolution of systemic states to answer this question (for a more detailed treatment, see pp. 86–88 in [1]). Recall from above, that the information length reflects the accumulated changes in probability densities as time progresses. If a system attains nonequilibrium steady state after a period of time, then the information length asymptotes to the distance between the initial (particular) state and the final (steady) state. This means that we can characterise a certain kind of particle (or creature) that returns to steady state in terms of the (critical) time τ it takes for the information length to stop increasing:(11)$$d\ell (\tau )\approx 0\Leftrightarrow D[q_{\tau }(\eta_{\tau },\pi_{\tau })||p(\eta_{\tau },\pi_{\tau })]\approx 0$$

The probability density $q_{\tau }(\eta_{\tau },\pi_{\tau })$ is the *predictive density* over hidden and sensory states, conditioned upon the initial state of the particle and subsequent trajectory of autonomous states. In brief, particles with a short critical time[note 18] will, effectively, converge to nonequilibrium steady-state quickly and show a simple self-organisation (e.g., the Aplysia gill and siphon withdrawal reflex) mathematically, these sorts of particles quickly ‘forget’ their initial conditions. Conversely, particles with a long critical time will exhibit itinerant density dynamics (e.g., you and me). Particles like you and me ‘remember’ our initial conditions and look as if we are pursuing long-term plans.

Convergence to nonequilibrium steady state in the future allows us to relate the surprisal of a trajectory of autonomous states (i.e., a policy) to the variational free energy expected under the predictive density above:(12)$$\begin{array}{cc} G(\alpha [\tau ]) & \approx A(\alpha [\tau ]|\pi_{0})\\ G(\alpha [\tau ]) & ≜\underset{energy}{\underset{︸}{E_{q_{\tau }}[I(\eta_{\tau },\pi_{\tau })]}}-\underset{entropy}{\underset{︸}{H[q_{\tau }(\eta_{\tau }|\pi_{\tau })]}}\\ & =\underset{ambiguity}{\underset{︸}{E_{q_{\tau }}[I(\pi_{\tau }|\eta_{\tau })]}}+\underset{risk}{\underset{︸}{D[q_{\tau }(\eta_{\tau }|\pi_{\tau })||p(\eta_{\tau })]}} \end{array}$$

The expected free energy in (12) has been formulated to emphasise the formal correspondence with variational free energy in (5): where the complexity and accuracy terms become *risk* (i.e., expected complexity) and *ambiguity* (i.e., expected inaccuracy). This path integral formulation says that if the probability density over systemic states has converged to nonequilibrium steady state after some critical time, then there can be no further increase in information length. At this point, the probability of an autonomous path into the future becomes the variational free energy the agent expects to encounter. 

The equality in (11) is a little abstract but has some clear homologues in stochastic thermodynamics (in the form of integral fluctuation theorems) [12,79]. Here, it tells us something rather interesting. It means that creatures that have an adaptive response to changes in their external milieu will look as if they are selecting their long-term actions on the basis of an expected free energy. Crucially, this free energy is based upon a generative model that must extend at least to a (critical) time in the future when nonequilibrium steady state is restored. Conversely, if certain kinds of creatures select their actions on the basis of minimising expected free energy, they will respond adaptively to changes in external states. 

This formulation offers a description of different kinds of particles or creatures quantified by their critical time or temporal depth in (11). For example, if a certain kind of particle (e.g., a trial or protozoan) has a short temporal horizon or information length, it will respond quickly and reflexively to any perturbations—for as long as it exists. Conversely, creatures like us (e.g., politicians and pontiffs) may be characterised by deep generative models that see far into the future; enabling a move from homoeostasis to allostasis and, effectively, the capacity to select courses of action that consider long term consequences [80,81,82]. Given that the imperative for this action selection is to minimise expected free energy (i.e., expected surprisal or uncertainty), we now have a plausible description of intentional behaviour that will, to all intents and purposes, look like uncertainty resolving, information seeking, epistemic foraging [26,81,83,84,85,86,87,88,89,90]. Alternatively, on a more (millennial) Gibsonian view, action selection responds to long-term epistemic affordances [91,92,93]. 

This temporal depth may distinguish between different kinds of sentient particles. Again, all of this follows in a relatively straightforward way from information theory and statistical physics. Furthermore, the equations above can be used to simulate perception and intentional behaviour. To illustrate the difference between short term (shallow) inference based upon Equation (5) and long-term (deep) active inference based upon Equation (12), we provide two examples in Figure 4 and Figure 5. The first uses simulated handwriting that is elicited purely on the basis of reflexive responses prescribed by a dynamic generative model (i.e., a pattern generator), while the second calls on the notion of epistemic affordance by simulating saccadic searches and active vision. In the present thesis, simulations of the second sort of active inference may offer a better account of intentional behaviour; namely, beliefs about the consequences of action and subsequent action selection.

## 9. Markovian Monism

Above, we have shown that a duality—between two ways in which states of a system can be conceived of—already arises at a very fundamental level; namely, for all systems that possess a Markov blanket. Their internal states can both be associated with an intrinsic and with an extrinsic information geometry. What metaphysical implication does this have? Does it follow that all systems with a Markov blanket have a mind (because they have probabilistic beliefs about external states)? Are such systems conscious? The formalism itself does not answer these questions: different metaphysical interpretations of the existence of a dual information geometry are possible. In fact, one might ask whether it has any metaphysical significance whatsoever. For the existence of an extrinsic information geometry only means that one *can* map internal states to conditional probability distributions (over external states, given blanket states). It does not mean that the resulting descriptions refer to entities that actually exist (just as we can ascribe to a lectern the propositional belief that the best way to persist is to do nothing—which does not mean that the lectern actually has a propositional belief; see [99]).

Hence, any metaphysical conclusions must be drawn with care. In what follows, we will first argue that the formalism speaks in favour of monistic views—if we assume that the existence of an extrinsic information geometry has any relevance for understanding the mind and consciousness in the first place. After that, we will discuss different interpretations of the dual perspective afforded by the two information geometries: panprotopsychism, neutral monism, dual-aspect theories, and physicalism. We will argue that physicalism provides the most plausible interpretation. However, we acknowledge that competing interpretations cannot conclusively be ruled out. Hence, we dub the resulting view ‘Markovian monism’. Markovian monism consists of two claims: (1) Fundamentally, there is only one type of thing and only one type of irreducible property (this is why it is a Markovian *monism*). (2) All systems possessing a Markov blanket have properties that are relevant for understanding the mind and consciousness: if such systems have mental properties, then they have them partly by virtue of possessing a Markov blanket (this is why it is a *Markovian* monism).

Why do we rule out dualistic interpretations of the dual information geometry? First, note that dualism is still consistent with the existence of an extrinsic information geometry. However, consider any properties that a system has by virtue of the fact that its internal states encode probability distributions over external states. Since the dynamics that can be described with reference to these properties can equivalently be described without regarding internal states as representations of probability distributions, there is a sense in which both perspectives are reducible to one another. Hence, the dual information geometry itself does not entail property dualism. Therefore, if one believes that there are irreducible mental properties, one has to posit them in addition to, and largely independently of, properties entailed by the existence of an extrinsic information geometry. But this means that mental properties will not be instantiated (partly) by virtue of the existence of a Markov blanket (contradicting claim (2) above). In other words, dualism is more or less orthogonal to the formal treatment.

However, we do believe that the existence of an extrinsic information geometry tells us something interesting about the origin of minds and consciousness. Under the assumption that properties entailed by the existence of a Markov blanket are relevant to understanding mental properties, we therefore have to reject dualism. This still leaves different metaphysical options open.

## 10. Markovian Monism as Panprotopsychism?

According to panpsychism, mental properties are fundamental non-physical properties and are instantiated by all micro-level entities. Hence, this amounts to a form of property dualism, which we ruled out above. Note, again, that dualism is compatible with the formal treatment presented here, but it would not be an interpretation in which properties entailed by the existence of a Markov blanket have any explanatory relevance to the existence of minds and consciousness—because panpsychism already presupposes mentality as a fundamental part of reality.

However, there is a variant of panpsychism, viz. panprotopsychism, that could, in principle, be described as a Markovian monism. In short, panprotopsychism is “the view that fundamental entities are *proto-conscious*, that is, that they have certain special properties that are precursors to consciousness and that can collectively constitute consciousness in larger systems.” [100], p. 259. These special, non-structural properties are *protophenomenal* properties that are not identical to (micro-)physical properties (otherwise, even physicalism could be considered as a form of panprotopsychism, [100], p. 260). There is nothing it is like to be a system that has just a single protophenomenal property. However, if a system displays a sufficiently large number of protophenomenal properties, or if they are arranged in the right way, then the system will also have phenomenal properties (which are constituted by collections of protophenomenal properties).

From the point of view of Markovian monism, one could identify properties entailed by the existence of a Markov blanket with protophenomenal properties. An example is the property of encoding a conditional probability distribution over external states. However, it is unclear to us to what extent this could be regarded as a non-structural property. Furthermore, a robust version of panprotopsychism would have to presuppose that all systems with a Markov blanket actually represent probability distributions—as opposed to just being systems that *can be described as if* they represented such distributions. Below, we will suggest that a realist interpretation of descriptions afforded by the extrinsic information geometry should be contingent on further conditions. This is why we would not interpret Markovian monism as a version of panprotopsychism.

## 11. Markovian Monism as Neutral Monism?

Neutral monism is normally read as a family of views; according to which the fundamental layer of reality consists of ontologically neutral entities. Different versions of the theory make different claims about the sense in which basic entities are neutral (see [101], who lists five different options). The most popular options seem to be views according to which the basic entities are (a) intrinsically *neither* mental nor physical or (b) intrinsically *both* mental and physical.

A great advantage of neutral monism is that it solves the mind-body problem without postulating two basic types of entity (mental and physical)—the significance of this is that worries about psycho-physical interaction (that plagued Cartesian dualism) disappear. The only causal interaction in question involves neutral entities (however, the problem of mental causation may reappear, in the sense that macro-level mental properties may still be causally irrelevant, see [102], pp. 33–34).

Markovian monism could be specified as a version of neutral monism in which basic entities are intrinsically neither mental nor physical. There are two conjugate ways in which things that exist can be described: either from the perspective of the intrinsic information geometry or from the perspective of the extrinsic information geometry. Under the assumption that neither perspective is privileged, one would have to conclude that reality is, fundamentally, ontologically neutral.

However, this would also presuppose a realist interpretation of descriptions in terms of the extrinsic information geometry (i.e., one would have to assume that all systems with a Markov blanket actually represent probability distributions and perform computations). Furthermore, it would have the consequence that even relatively simply systems, such as single-cell organisms, would have a mind (as suggested by [103]). For these reasons, we would not interpret Markovian monism as a version of neutral monism.

## 12. Markovian Monism as a Dual-Aspect Theory?

Dual-aspect monism is the position that reality has two aspects: a mental and a physical aspect. Dual-aspect monism is very similar to neutral monism. Depending on how it is defined, it may even collapse into neutral monism (or into panpsychism, see [104], p. 366). For instance, if dual-aspect monism is defined as the view that reality is, at a fundamental level, both physical and mental, then this comes extremely close to the view that basic entities are intrinsically both mental and physical—and hence to a version of neutral monism (see [101], sec. 8.3).

Furthermore, if the *aspect* in ‘dual-aspect’ is interpreted in terms of properties, such that basic entities have both mental and physical properties (as suggested by [105], p. 46), then dual-aspect theory becomes a form of property dualism—which we ruled out above. 

There are versions of dual-aspect theory that explicitly refrain from defining the dual aspect in terms of property dualism (see, e.g., [106], pp. 339,342). Markovian monism is similar to dual-aspect monism (cf. [107], pp. 220–221), in that it entails that one and the same thing (i.e., internal states of a system possessing a Markov blanket) can be viewed from two perspectives. Internal states can either be regarded as states of a random dynamical system; or they can be viewed as the parameters of a probability distribution (i.e., probabilistic belief). In order to count as a dual-aspect monism, these two perspectives would have to be mutually irreducible (see [106], p. 46; [105], p. 341), we are sceptical that this would be a coherent interpretation of the dual information geometry.

As with the other two interpretations discussed above, an interpretation in terms of a dual-aspect monism would presuppose a realist view on descriptions in terms of the extrinsic information geometry. Furthermore, just as the interpretation in terms of neutral monism, it would entail that single-cell organisms have a mind. In what follows, we will sketch how Markovian monism can ground versions of reductive materialism. This physicalist interpretation of Markovian monism is the one we favour—although we admit that other interpretations cannot conclusively be ruled out.

## 13. Markovian Monism as Reductive Materialism

Here is what we believe is the most coherent interpretation of the formal treatment. The fact that one can associate two information geometries with systems possessing a Markov blanket reveals a continuity between simple, non-conscious systems and more complex, conscious systems such as human beings: due to the extrinsic information geometry, simple systems can be described *as if* they had beliefs about external states. For conscious systems, the perspective—afforded by such ‘as if’ descriptions—acquires a special status, because it will typically abstract away from many of the details inherent in the mechanistic perspective. For instance, the probabilistic beliefs ascribed to explain cognitive phenomena are typically assumed to be represented by the average activities of neuronal *populations*, which means that any differences between populations with the same average properties will be irrelevant from the perspective of the extrinsic information geometry. This squares well with the idea that causation (including mental causation) is a macroscopic phenomenon [108,109]. At the same time, these macrostates are always grounded in more fine-grained physical states, and their properties can be reductively explained in terms of physical properties.

Furthermore, the computational properties ascribed to conscious systems will be more numerous and more complex than those ascribed to non-conscious systems. There are no additional, non-reducible properties, which are necessary to explain the mind and consciousness; between some non-conscious and conscious systems, there is only a gradual difference. This entails that consciousness is a vague concept; i.e., there will be borderline cases in which the concept cannot unequivocally be applied.

In particular, this proposal rests upon a distinction between temporally deep and shallow generative models, that accompanies the distinction between conscious and unconscious inference. This distinction is vague, in the sense that any generative model of dynamics has, to a certain extent, temporal depth. For example, predictive coding, homoeostasis and thermostats can all be articulated in terms of perceptual control [46,110,111] and a reflexive form of active inference using generative models based upon differential equations. The fact that a generative model entails differential equations means that there is some inference over time. The distinction between deep and shallow then becomes a quantitative issue. Perhaps a better distinction would be between generative models that entertain a single trajectory into the future, versus multiple (counterfactual action dependent) trajectories that incur a selection problem; namely, choosing an action or planning.

Construing consciousness as a vague concept may even have relevance for the meta-problem of consciousness [112]; i.e., the problem of explaining why it seems (to many) that a physical duplicate of a conscious creature could be non-conscious. Although solving the meta-problem is not the aim of this paper,[note 19] we can at least contribute to an explanation: as noted above, our interpretation of Markovian monism entails that there is only a gradual difference between some non-conscious and conscious systems, and that consciousness is a vague concept. So, when people claim they can imagine a physical duplicate that is unconscious, they may in fact imagine not a complete duplicate, but a system that differs in (seemingly non-significant ways) from a conscious system. As an analogy, consider a heap of sand. A heap of sand is constituted by grains of sand. But, one could object, a heap of sand cannot be *just* a collection of grains of sand, because I can imagine a collection of grains of sand (say, three grains) that does not count as a heap. Hence, there seems to be a crucial difference between collections of grains of sand and *heaps* of sand—just adding a grain of sand to something that is not a heap does not turn it into a heap. Similarly, just adding a bit more structure and function to a non-conscious system does not turn it into a conscious system. Hence, it would seem as if consciousness requires more than just the right structure and functions, and the hard problem arises. But if consciousness is a vague concept (as suggested by our interpretation of Markovian monism), then the right structure and functions can be metaphysically sufficient for consciousness, even if adding just a bit of structure and function to any uncontroversially non-conscious system does not make it conscious.

Furthermore, the very existence of the meta-problem implies a certain kind of Bayesian belief that entails some puzzlement about ‘our capacity to have subjective experiences of a quantitative sort’. But ‘qualia’ and accompanying ‘puzzlement’ are just Bayesian beliefs that imply an extrinsic information geometry. So, is there anything special about Bayesian beliefs about Bayesian beliefs? The answer is yes: beliefs about beliefs (in a mathematical sense) require a hierarchical generative model. But, a hierarchical generative model requires hierarchically deployed Markov blankets to introduce the necessary conditional independencies (which make it hierarchical). We therefore conclude that phenomenally conscious systems for which a hard problem exists must possess a certain kind of statistical structure; namely, Markov blankets within Markov blankets [4].

Although we believe that there are only gradual differences between non-conscious and conscious systems, if one merely considers the probabilistic beliefs that can be ascribed to such systems, there are still categorical differences that can be described in terms of more high-level properties, such as intentionality and computation (note that this does not imply a “phase transition” between unconscious and conscious systems).

In particular, one can make a threefold distinction between (i) systems that behave only ‘as if’ they implemented computations over probabilistic beliefs, (ii) systems for which the “computational stance” [114] provides added explanatory value, and (iii) systems that can usefully be described as not only computational, but also as representational systems.[note 20] While not speaking against a continuity between life and mind [103], this threefold distinction could be used to establish a discontinuity between life and consciousness.

Specifying the difference between (i) and (ii) would require defending a particular account of computation, which is beyond the scope of this paper. The step from (ii) to (iii)—i.e., from a computational to a representational system—requires ascribing content to internal states of the system. Representationalist interpretations of the free-energy principle refer to computations that are implemented (or approximated) by systems that minimize free energy (see, e.g., [117], pp. 571–572). Such computations are defined with respect to exactly the types of probabilistic beliefs encoded by systems with an extrinsic information geometry.

Although Markovian monism, interpreted as a form of reductive materialism, is not a theory of consciousness, it refers to properties that may ground mental properties (including phenomenal properties). As such, it provides a foundation for various physicalist approaches to consciousness and the mind, most notably representationalism and (computational) functionalism.[note 21]


## 14. Consciousness and Integrated Information

There have been previous attempts to use information theory to describe conscious processing. Perhaps the most notable is integrated information theory [118,119]. One might ask about the relationship between the free energy principle (FEP) and integrated information theory (IIT)? At the time of writing, there is a gap between these theoretical approaches. First, the FEP is a ‘first principle’ account that uses variational principles to build upon the Langevin formulation of random dynamical systems. In contrast, IIT is an ‘axiomatic’ approach that starts with some assumptions about what information processing must look like to be a contender for explaining conscious experience. The formal distinction between the FEP and IIT is that the free energy principle is articulated in terms of probabilistic beliefs *about* some (external) thing, while integrated information theory deals with probability distributions *over* the states of some system. In other words, IIT does not commit to an extrinsic information geometry (the “geometry of integrated information” is an intrinsic information geometry, see [120]). This is not necessarily a problem, in so far as IIT offers a normative (i.e., measurable, in principle) description of systems that comply with axioms, which inherit pre-theoretical notions of consciousness. On the other hand, both the free energy FEP and IIT can be cast in terms of information theory and in particular functionals (e.g., variational free energy and ‘phi’). Furthermore, they both rest upon partitions (e.g., Markov blankets that separate internal from external states and complexes that constitute conscious entities and can be distinguished from other entities). This speaks to the possibility of, at least, numerical analyses that show that minimising variational free energy maximises ‘phi’ and *vice versa*.

Although integrated information theory does not commit to a Markovian information geometry of experience (i.e., conscious or unconscious inference about something), it is possible to establish some kind of construct validity between the FEP and IIT in terms of the axioms upon which IIT is predicated. In other words, one can establish—at least heuristically—that the FEP features the essential properties of experience that constitute the axiomatic basis of IIT. There are five axioms; namely, intrinsic existence, composition, information, integration and exclusion. In brief:

**Intrinsic existence**—consciousness exists: each experience is actual and exists from its own intrinsic perspective. This is a necessary consequence of Bayesian mechanics under the free energy principle because the dynamics underlying inference are physically realised and are, by construction, intrinsic in the sense of pertaining to internal states.**Composition**—consciousness is structured: with multiple phenomenal distinctions. Again, this is a necessary aspect of Bayesian mechanics, which is defined in terms of the structure implicit in conditional independencies. Indeed, from a statistical perspective, minimising variational free energy is synonymous with structure learning [59,121,122].**Information**—consciousness is unique: each experience is the particular way it is, thereby differing from other possible experiences (i.e., differentiation). Again, this is a fundament of Bayesian mechanics under the free energy principle; in the sense that any information geometry implies a particular point on a statistical manifold (of internal or intrinsic states) maps to a particular probability or belief state with phenomenal support (i.e., an extrinsic belief distribution over the external states).**Integration**—consciousness is unified: each experience is irreducible to disjoint subsets of phenomenal distinctions (i.e., integration). Again, this is a necessary aspect of the information geometry that underwrites the free energy principle. This follows because for each point on the internal statistical manifold, there is a single probabilistic belief (i.e., variational density). In other words, although this density could be very high dimensional, it is just one probabilistic belief that cannot be dissembled or reduced. Another aspect of the axiom of integration is that “every part of the system has both causes and effects within the rest of the system” ([123], p. 3). This is true for systems possessing a Markov blanket, because the gradient flows of internal states (and associated belief updating) are, by definition, conditionally dependent.**Exclusion**—consciousness is definite: each experience is characterised by what it is (neither less no more than) and flows at the speed it flows (neither faster nor slower). Again, this is a necessary consequence of the density dynamics that underwrites the free energy principle. In other words, flows on the extrinsic (statistical) manifold are unique and entail particular probabilistic beliefs about external states, i.e., precise beliefs about being in a particular (external) state but not another. Furthermore, each probabilistic belief has its own sufficient statistics that exclude the possibility of other sufficient statistics. For example, beliefs about my temperature can be stipulated with an expectation that my temperature is such and such. This precludes the possibility that I expect to my temperature to be anything else. In contrast to the exclusion axiom, however, the existence of a Markov blanket at one spatiotemporal scale does not exclude the existence of (e.g., nested) Markov blankets at other spatiotemporal scales. 

In summary, on an informal review, the information geometry and density dynamics implied by Markov blankets appear to possess the qualities—or conform to the essential criteria—that constitute the axiomatic basis of integrated information theory.

The important result of this section, from our perspective, is that at least some properties associated with consciousness are already entailed by Bayesian mechanics under the free energy principle. This supports the (speculative) hypothesis that adding further constraints on generative models—entailed by systems possessing a Markov blanket—might enable us to say which systems are conscious, and which are not. Unconscious systems do not perform active inference in a way that entails that characteristic features of consciousness are instantiated, whereas conscious systems do. Specifying the constraints on generative models that underpin active inference of the sort that entails characteristic features of consciousness can lead to a unitary concept of consciousness ([124], as opposed to a bundle of feature descriptions; see [125]). In other words, a sufficiently specified sort of active inference may describe computational processes that account for clusters of features that are characteristic for consciousness—and thereby show *why* these features cluster together (cf. the natural kind approach sketched in [126], p. 7).

## 15. Information Geometry and Altered States of Consciousness

To recap, the information geometry above—and attending free energy principle—rest upon a separation of external from internal states by blanket states. This move is crucial for elaborating a physics of sentience, in which physical dynamics entail probabilistic beliefs about something. In this sense, it takes us beyond existing formalisms in the physical and philosophical sciences—revealing some key issues. For example, quantum treatments generally rely upon some specification of a Schrödinger potential. But where did this potential come from? Similarly, for statistical thermodynamics, where did the ‘heat bath’ (i.e., thermal reservoir) come from and what contains the heat bath? In short, there would be no quantum or statistical mechanics in the absence of Markov blankets (i.e., Schrödinger potentials and heat baths). The same questions can be posed to things like integrated information theory: what is this information about, in the absence of a Markovian (belief-based) information geometry? What principles explain the emergence and maintenance of partitions induced by complexes? 

The point here is that a Markovian monism (or information geometry) necessarily requires some notion of duality or conjugacy, here afforded by the Markov blanket. On this reading of self-evidencing to nonequilibrium steady state, some pressing questions arise. For example, what would happen if internal and external states were statistically sequestered. In other words, is there a sentient physics for isolated systems, such as those considered in statistical mechanics. From a neurobiological perspective, this speaks to altered states of consciousness that ensue with physiological or pharmacological quenching of blanket states. There are many examples that we could pursue here; including states of consciousness associated with psychedelic and psychomimetic drugs, or, indeed, the false inference associated with psychopathology (e.g., hallucinations and delusions). However, we will focus on a canonical example; namely, sleep and dreaming. So, what does sleep physiology tell us about conscious or unconscious inference?

If, for simplicity, we assume that the state of sleep corresponds to a sequestering of internal states from blanket states, we have an interesting preparation of a neuronal system that is temporarily—and repeatedly—isolated from the sensorium. This simplification is easily substantiated by many neurophysiological and neurochemical aspects of sleep physiology [127]. For us, the key question is: what happens to the Markovian information geometry and Bayesian mechanics of the internal (neuronal) states? At first glance, the notion of self-evidencing as an explanation for internal dynamics simply goes away. This is because the Lyapunov or potential function driving dynamics ceases to exist in the absence of blanket states (technically, the gradients that underwrite gradient flows disappear). However, at nonequilibrium steady-state, periods of disconnection from blanket states must themselves be transient and repetitive; i.e., be part of the itinerant dynamics that have a pullback attractor. This means Bayesian mechanics must still apply, even during the suspension of any coupling with blanket states. We will consider a physiological case (of sleep) in which in autonomous dynamics are still in play. In this setting, the variational free energy gradients are driven by the part of free energy that does not depend upon blanket states. This part is the complexity term of Equation (5), where removing blanket states discloses a description of complexity (i.e., redundancy) resolving internal dynamics:(13)$$\begin{array}{c} f_{\mu }(\mu )\approx (Q_{\mu \mu }-\Gamma_{\mu \mu })\nabla_{\mu }F(\mu )\\ F(\mu )=\underset{complexity}{\underset{︸}{D[q_{\mu }(\eta )||p(\eta )]}} \end{array}$$

In other words, neuronal dynamics during sleep will appear to minimise the complexity of the generative model (i.e., minimise the divergence between the posterior beliefs and prior beliefs—in the absence of sensory evidence). This is precisely the argument put forward in statistics for optimising models in the absence of new statistical data—by removing redundant model parameters [128]. In neurophysiology, this is the argument that we have made previously to explain the very existence of sleep phenomenology—and in particular, the role of dreaming [129,130,131]. In short, physiological states of altered consciousness, such as sleep, may offer an important empirical handle on theoretical notions—notions that arise from the variational principles of sentience.

In summary, an extrinsic information geometry can exist in the (temporary) absence of blanket states, in virtue of prior beliefs held by internal states. These prior beliefs underwrite proto-consciousness [127] and are necessary to generate virtual or fictive realities [132] in states such as dreaming [129,130,131]. There are many fascinating issues here; for example, the complexity term of the free energy functional above provides a compelling metaphor for the housekeeping that we may enjoy during sleep [71,133]. This complexity minimisation itself has formal links with both machine learning [128] and universal computation [88,90,134]—and physiology in the form of synaptic homoeostasis [133,135].

## 16. Conclusions

In conclusion, we have rehearsed some of the cornerstones of statistical physics and information theory to show how the very existence of things (i.e., Markov blankets) necessarily induces an information geometry with two aspects. First, the dynamics of physical (internal) states of any sentient particle or creature is equipped with an information geometry, in terms of time dependent changes in probability distributions over internal states. We have called this an *intrinsic* information geometry. At the same time, there is a conjugate information geometry, which pertains to probability densities over external states parameterised by internal states. We have called this an *extrinsic* information geometry (because it is predicated upon probabilistic beliefs about external states). Crucially, the two are formally and fundamentally linked—in that the dynamics of internal states can always be expressed as a gradient flow on a variational free energy functional of belief (protophenomenal) states. This construction is entirely consistent with forces cast in terms of stochastic thermodynamics, with the appropriate constant of proportionality (i.e., Boltzmann’s constant and the temperature). 

Second, we have considered the time it takes for a particle or creature to return to its attracting manifold (i.e., nonequilibrium steady state) from an initial state. When treated in the form of a path integral or fluctuation theorem, this temporal aspect may distinguish among different kinds of creatures; depending on how deeply their generative model (entailed by internal states) considers the future; c.f., counterfactual depth [111,136]. This is functionally equivalent to the temporal depth or extent of policies; namely, courses of action, and internally consistent with the notion of planning as inference [85,87,137].

Another technical formulation of information-processing—that is closely related to information geometry—is the use of gauge theories (e.g., the celebrated theory of general relativity). Our own work in this area [138] focused on gauge theories associated with information geometry and the Fisher information metric. Recall that the Fisher information metric that equips the belief space or statistical manifold (here, afforded by internal states) with a geometry has a number of revealing interpretations. First, the Fisher information metric is simply the curvature of the variational free energy as one moves on the internal (statistical) manifold. This is the same as the conditional precision or confidence placed in beliefs about external states. From a psychological perspective, this curvature or precision plays a key role in predictive processing (i.e., Bayesian brain) accounts of attentional selection and, a particularly important role in interoceptive inference [67,68,76,139,140]. We emphasise this seamless connection from gauge theories—through information geometry and variational inference—to precision for a special reason. The central role of precision and confidence in mediating consciousness is exactly the endpoint of the phenomenological and neuropsychological analysis of conscious processing and selfhood offered by Mark Solms [107]. Furthermore, the ‘paper trail’ from gauge theory to attention endorses pre-theoretical notions about their intimate relationship [141,142]. One could develop this story even further, in terms of the predictive processing of precision *per se*—and how this may underwrite mental action and a sense of agency [111,139].

In terms of the philosophy of science, perhaps the most tenable way of treating a dual aspect information geometry is under structural realism. We mean this in the sense that the mathematical and geometric form (i.e., structure)—afforded by the mathematical analysis above—allows one to say something about the relationship between (probabilistic) beliefs and the (statistical) physics of internal states that ‘hold’ or ‘represent’ those beliefs. Structural realism takes the pressure off any strong ontological commitments to the mapping between information structures and their content. However, this information structure implies a lawful dependency of probabilistic beliefs (about external states) and parameterised probability distributions (over internal states), in the following sense. Any movement on the internal statistical manifold will necessarily be accompanied by a movement in belief space, as measured by the information length or distance between the beliefs that are parameterised by expected internal states. Furthermore, because these internal states lie upon a statistical manifold of conditional expectations, they must play the role of thermodynamic variables. It follows that belief updating and statistical thermodynamics both supervene on the same internal manifold. Note, the claim here is that physics (i.e., statistical thermodynamics) supervenes on the same statistical manifold as belief updating. This supervenience—on the same statistical manifold—from which both information geometries inherit their structure could be read as the philosophical formulation of the mathematical conjugacy implied by intrinsic and extrinsic information geometries.

In terms of the ontological commitments beyond this structural (realism) argument, any claims would have to be argued much more carefully. It is tenable to associate physics (in the sense of quantum, statistical and classical) mechanics with the intrinsic information geometry. Indeed, this is common parlance in statistical physics [62,64,143]. The more delicate issues arise in terms of commitments to—or interpretation of—the second (extrinsic) sort of information geometry that underwrites Bayesian mechanics. One can avoid any strong ontological commitments here and simply note that should there be any philosophical sentience (i.e., ‘qualia’) in play, they are more likely to be an attribute of belief updating—and therefore part of Bayesian mechanics. We have approached this issue by suggesting Markovian monism entails a gradual difference between non-conscious and conscious entities, and—in this sense—consciousness is a vague concept.

## Figures and Tables

**Figure 1 entropy-22-00516-f001:**
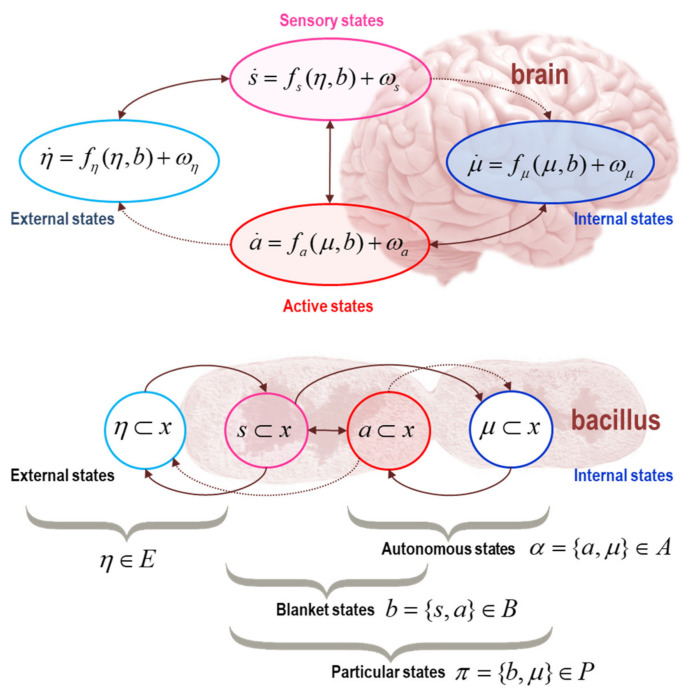
(*Markov blankets*): This schematic illustrates the partition of systemic states into internal states (blue) and hidden or external states (cyan) that are separated by a Markov blanket—comprising sensory (magenta) and active states (red). The upper panel shows this partition as it would be applied to action and perception in the brain. The ensuing self-organisation of internal states then corresponds to perception, while action couples brain states back to external states. The lower panel shows the same dependencies but rearranged so that the internal states are associated with the intracellular states of a Bacillus, while the sensory states become the surface states or cell membrane overlying active states (e.g., the actin filaments of the cytoskeleton).

**Figure 2 entropy-22-00516-f002:**
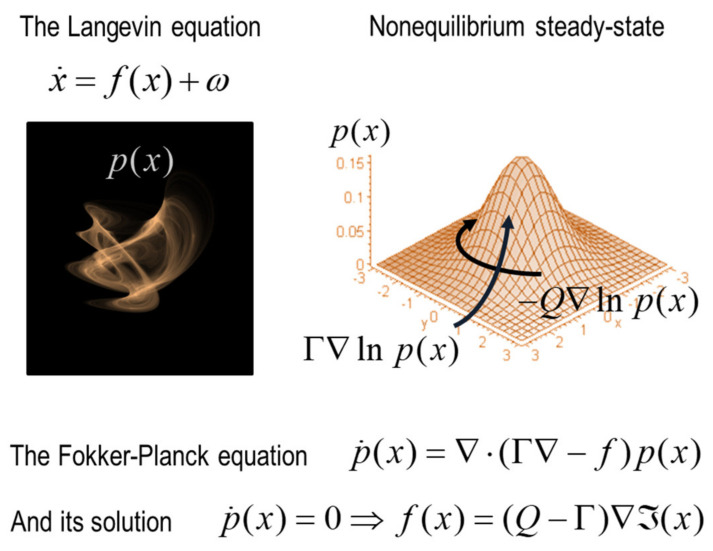
(*density dynamics and pullback attractors*): This figure illustrates the fundaments of density or ensemble dynamics in random dynamical systems—of the sort described by the Langevin equation. The left panel pictures some arbitrary random attractor (a.k.a., a pullback attractor) that can be thought of in two ways: first, it can be considered as the trajectory of (two) systemic states as they evolve over time. For example, these two states could be the depolarisation and current of a nerve cell, over several minutes. At a larger timescale, this trajectory could reflect your daily routine, getting up in the morning, having a cup of coffee, going to work and so on. It could also represent the slow fluctuations in two meteorological states over the period of a year. The key aspect of this trajectory is that it will—after itinerant wandering and a sufficient period of time—revisit particular regimes of state space. These states constitute the attracting set or pullback attractor. The second interpretation is of a probability density over the states that the system will be found in, when sampled at random. The evolution of the probability density is described by the Fokker-Planck equation. Crucially, when any system has attained nonequilibrium steady state, we know that this density does not change with time. This affords the solution to the Fokker-Planck equation—a solution that means that there is a lawful relationship between the flow of states at any point in state space and the probability density. This solution expresses the flow in terms of gradients of log density or surprisal and the amplitude of random fluctuations. In turn, the nonequilibrium steady-state solution can always be expressed, via the Helmholtz decomposition, in terms of two orthogonal components. One component is a gradient flow that rebuilds probability gradients in a way that is exactly countered by the dispersion of states due to random fluctuations. The other component is a solenoidal or divergence-free flow that circulates on isoprobability contours. These two components are shown in the schematic on the right, in terms of a curl-free gradient flow—that depends only on the amplitude of random fluctuations Γ– and a divergence-free solenoidal flow—that depends upon an antisymmetric matrix *Q*. This example shows the flow around the peak of a probability density, with a Gaussian form. Please see [1,25] for details.

**Figure 4 entropy-22-00516-f004:**
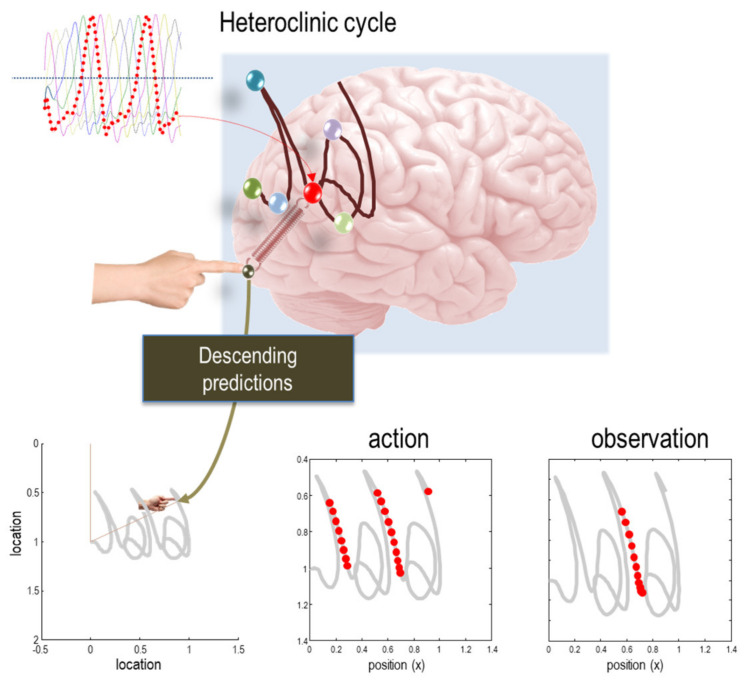
(*autonomous movement*). This figure shows the results of simulating active inference (i.e., writing), in terms of conditional expectations about hidden states of the world, consequent predictions about sensory input and the ensuing behaviour. The autonomous dynamics that underlie this behaviour rest upon prior expectations about states with Lotka-Volterra dynamics (c.f., a central pattern generator): these are the six (arbitrarily) coloured lines in the upper left panel. In this generative model, each state is associated with a location in Euclidean space that attracts the agent’s finger. In effect, the internal states then supply predictions of what sensory states should register, if these prior beliefs were true. Active states try to suppress the ensuing prediction error (i.e., sensory surprisal) by reflexively fulfilling expected changes in angular velocity, through exerting forces on the agent’s joints (not shown). The subsequent movement of the arm is traced out in the lower left panel. This trajectory has been plotted in a moving frame of reference, so that it looks like synthetic handwriting (e.g., a succession of ‘j’ and ‘a’ letters). The lower left panels show the activity of one (the fourth attractor) conditional expectation under ‘action’, and ‘action-observation’. During action, sensory states register both the visual and proprioceptive consequences of movement, while under action observation, only visual sensations are available—as if the agent was watching another agent. The red dots correspond to the time bins during which this state exceeded an amplitude threshold of two arbitrary units. They key thing to note here is that this unit responds preferentially when, and only when, the motor trajectory produces a down-stroke, but not an up-stroke. Please see [94] for further details. Furthermore, with a slight delay, this internal state responds during action and action observation. From a biological perspective, this is interesting because it speaks to an empirical phenomena known as mirror neuron activity [95,96,97].

**Figure 5 entropy-22-00516-f005:**
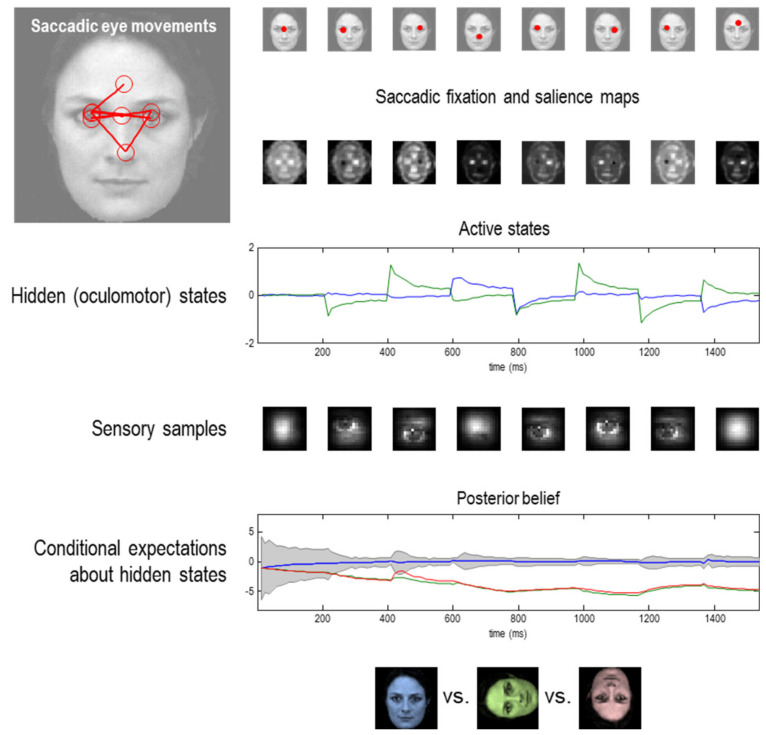
(*epistemic foraging*). This figure shows the results of a simulation in which a face was presented to an agent, whose responses were simulated by selecting active states that minimise expected free energy following an eye movement. The agent had three internal images or hypotheses about the stimuli she might sample (an upright face, and inverted face and a rotated face). The agent was presented with an upright face and her conditional expectations were evaluated over 16 (12 ms.) time bins, until the next saccade was emitted. This was repeated for eight saccades. The ensuing eye movements are shown as red dots at the end of each saccade in the upper row. The corresponding sequence of eye movements is shown in the insert on the upper left, where the red circles correspond roughly to the proportion of the visual image sampled. These saccades are driven by prior beliefs about the direction of gaze based upon the saliency maps in the second row. These saliency maps are the expected free energy as a function of policies; namely, where to look next. Note that these maps change with successive saccades as posterior beliefs about external states, including the stimulus, become progressively more precise or confident. Note also that salience is depleted in locations that were foveated in the previous saccade because these locations no longer have an epistemic affordance (i.e., the ability to reduce uncertainty or expected free energy). This is a nice illustration of a ubiquitous phenomenon, known as inhibition of return. Oculomotor responses are shown in the third row in terms of the two hidden oculomotor states, corresponding to vertical and horizontal eye movements. The associated portions of the image sampled (at the end of each saccade) are shown in the fourth row. The final two rows show the posterior beliefs in terms of their sufficient statistics and the stimulus categories, respectively. The posterior beliefs are plotted here in terms of conditional expectations and the 90% confidence interval about the true stimulus. The key thing to note here is that the expectation about the true stimulus supervenes over its competing expectations and, as a result, conditional confidence about the stimulus category increases (the confidence intervals shrink to the expectation). This illustrates the nature of evidence accumulation when selecting a hypothesis or percept the best explains sensory data. Within saccade accumulation is evident even during the initial fixation with further stepwise decreases in uncertainty as salient information is sampled at successive saccades. Please see [98] for further details.

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
