# Peer review of "Sentience and the Origins of Consciousness: From Cartesian Duality to Markovian Monism"

_entropy, 2020, doi:10.3390/e22050516_

Round 1

Reviewer 1 Report

This paper proposed a novel view on consciousness, i.e., Markovian monism. The authors pointed out that if there is a Markov blanket in a system, the variables in the system can be naturally partitioned into internal and external states, which are conditionally independent given blanket states, and the dynamics of internal states can be interpreted as a gradient flow on variational free energy, which is a functional of probabilistic beliefs about external states. This interpretation per se does not necessarily explain consciousness but the authors indicated that we may be able to associate ‘mental’ states with probabilistic beliefs encoded by internal states and interpret this as the duality between physical and phenomenal properties.

I think this paper is extremely interesting, valuable, and inspiring for scientific studies of consciousness. To be honest, I have not yet understood the technical details of the paper at this point but I will definitely follow all the technical details after I submit this report and contemplate more its relation to consciousness. Having said that, I would like the authors to clarify the following point regarding the relationship between the free energy principle (FEP) and IIT.

The authors compared partitions provided by Markov blankets with Minimum Information Partitions (MIPs) in IIT. However, the corresponding notion in IIT is not MIP but “complex” (Balduzzi & Tononi, 2008 , PLoS Comp Biol; Oizumi et al., 2014, PLoS Comp Biol). A complex is a locally most integrated subsystem where integrated information is larger than its subsets and supersets (integrated information is evaluated across MIP in a system). IIT hypothesizes a complex as the locus of consciousness. This means that the variables within a complex are internal states, which directly influences its conscious states, and the variables outside of the complex are external states, which only indirectly influence its conscious states and are considered as background conditions. There could be multiple complexes in a system and each complex constitutes a conscious entity. In this way, complexes determine the boundaries between multiple conscious entities such as the boundaries between two different persons or the boundaries between the left and right brain in split-brain patients.

I think that Markov blankets and complexes are similar concepts, which both determine the boundaries of consciousness, and thus, it is interesting to specify a difference between them. For example, if we consider a fully-connected network with all-to-all reciprocal connectivity, there is no Markov blankets in the system. Thus, in this case, there cannot be any distinctions between internal and external variables and it is unclear how we can interpret consciousness that can be generated in this system based on Markovian monism.

On the other hand, we can always determine the boundaries based on complexes. As a simple illustrating example, consider a system consisting of two subsystems A and B. Again, assume that all the elements in the whole system (A+B) are reciprocally connected. Consider the case where the inner connections within each subsystem A and B are much stronger than the inter connections across the subsystems A and B. In this situation, IIT would predict subsystem A and B are both complexes and the whole system (A+B) is not a complex because integrated information within each subsystem A and B would be larger than integrated information within the whole system (A+B). From the perspective of subsystem A, the variables within subsystem B are external states and from the perspective of subsystem B, the variables within subsystem A are external states. This is IIT’s explanation why split-brain occurs when the connections between the left and right brain are weakened (Tononi, 2008, Biol Bull). If we apply the framework of Markov blankets to this split-brain example, I think we cannot find any partitions among variables because all of the variables in the system are reciprocally connected and even the variables within subsystem A and those within subsystem B are weakly connected.

I am interested in how the framework of Markovian monism would interpret the above-mentioned case of split-brain for example. I understand that the Markovian monism itself is not designed for explaining consciousness and thus, may require additional principles for explaining this phenomenon, i.e., split of consciousness. The authors seem to concentrate on the case where there are Markov blankets but I do not see a strong reason why we only need to consider such cases when we discuss consciousness. I would like to know how this theoretical framework can be utilized in more general cases, where there is no clear Markov blankets.

Minor

l.982 “Our own work in this area (XXX) …”

It seems like a citation error here.

l.1030 !!! INVALID CITATION !!!

Please correct this.

Reviewer 2 Report

The manuscript demonstrates how the dynamics of a temporally sustained system can be interpreted in (at least) two different ways: as probabilistic “beliefs” about the external environment of the system and as the temporal evolution of the system’s internal states. Subsequently, the authors propose that the former interpretation could account for “mental” properties assigned to the system, or even “qualia”. In this account, the mental properties are tied to the dynamical (also termed “physical”) properties and thus can be reduced to the physical properties. This view is thus termed “markovian monism” and categorized as a form of reductive materialism.

In principle, the paper could be published almost as it (fixing the broken refs etc and addressing point (2) below). Nevertheless, I have opted for “major revisions” as I also believe the manuscript could be considerably improved (see points below). In addition, I have taken the opportunity to pose several questions regarding the theoretical framework and the logic of the proposed argument. Clarifying these issues could also help strengthen the argument put forward by the authors.

Main Issues:

  1. The paper is very long. The first part (p. 2-16) is hard to read and mostly has been presented before (and personally I understood just as much or as little as in previous overviews of the Free Energy Principle). It could improve the paper substantially if only the main logic structure of the argument were laid out, instead of providing the whole derivation of all quantities, those could simply be referenced (see below).
  2. There are some inconsistencies in the presentation (see below).

(1) Here I will outline the authors’ argument how I understood it with some questions regarding the theoretical formalism. In general I would recommend cutting down on all peripheral associations that are very well interesting in a larger context but do not contribute to the actual line of reasoning necessary for making the case for Markovian Monism (e.g., L 129-138; not sure 193-211 is really necessary and most is repeated in the following; also L. 256-276 doesn’t seem directly relevant; L. 321-326; L. 423-435;…)

a) Markov Blankets separate internal from external states.

Q: Basically everything has a Markov Blanket, correct? That includes feedforward systems, and also sets of multiple non-interacting systems.

b) Some systems (those that are stable over some time) have, or can be characterized as having, an attracting set. Non-trivially this should be a nonequilibrium steady-state. These systems follow gradients. These systems are said to “exist”.

Q: What is the relevant time scale for the attractor for human consciousness? It seems from the caption of Fig. 2 and the sleep example at the end that the relevant scale is on the order of a life-time. Is that right?

Q: I’m having some difficulties squaring the notion of stationary probabilities and an attracting set with open/living systems. Do all the equations still hold when the system is on a transients, or only when it is on the attractor?

c) Surprisal minimization refers to particular states (blanket + internal)

Separating the blanket from the internal states we can divide the surprisal minimization up into perception and action which then seem to work together to minimize surprisal. Variational free energy is sort of equivalent with surprisal, but easier to deal with (is that really relevant here?).

d) Beliefs: An example would be good. For instance, a bacterium could have the belief that … because … Since beliefs are later associated with qualia it needs to be made very clear what is and is not entailed by the word in this article (see point 2 below). If I understand it correctly, it is just conditional probabilities. E.g.: given that my internal state is x, this constrains the possibilities of external states. Nothing more, correct? Can we say that a “V1 neuron firing specifies a belief about a visual stimulus”? Or what would be an example?

e) Information geometry: We can infer probability distributions about external states from the internal states. These probability distributions form one information geometry (which means I can measure distances between the distributions etc.). At the same time I can describe the internal dynamics based on conditional probabilities (future states given current states), which also make for an information geometry in that I can measure distances between the distributions. Yes?

Q: Why would the internal metric not correspond to beliefs (of the system about its future states)? I.e. why are only the extrinsic inferences “beliefs”?

Q: It seems to me that the intrinsic and extrinsic information geometries are not qualitatively different and calling one physical and the other computational is just another way to say one is about the system’s own states whereas the other is about external states. Am I missing something? Moreover, while the extrinsic information geometry is reducible to the intrinsic information geometry the same is true the other way around (mathematically), no? Why should the physical take precedence? (I guess this is discussed to some extend in l. 574-582)

f) Active inference
Q: Now it seems the external states matter while they did not before? Is this section necessary? Which systems have generative models? All systems that exist (see b)?

g) Active inference and information length/temporal depth: The point relevant to the later sections seems l. 502. Maybe the rest can be cut down.

(2) Inconsistencies:

  • 78-82: Here the authors state that they will stay away from phenomenology and qualia. Yet most of the discussion starting on p. 16 makes very strong claims about the nature of qualia as “beliefs” in the Bayesian sense. See also l. 362-363.

  • The distinction between unconscious and conscious systems (l. 583-600) does not follow from the formalism. If I understood correctly, basically everything is equipped with an extrinsic (and intrinsic) geometry). If consciousness is equated with the extrinsic geometry it just means it can be graded. Instead, now arbitrary functional criteria are imposed to distinguish presumably conscious from presumably unconscious systems. E.g., who says that single cell organisms are not conscious (as in “it feels like something to be a single cellular organism, even if it doesn’t feel like much”) (l. 633)?
    I also understand the heap analogy below, however, I would argue that phenomenology as the extrinsic geometry is not the “heap” in the example but simply the “sand”. For certain purposes, i.e. to be called a “heap” you need enough sand, but not to have “sandiness” per se.
  • 620-630: This only pertains if you require higher-order thoughts as a necessary property for phenomenology and many would disagree. This implicit assumption needs to be spelled out at least. Note again, it does not follow from the mathematical formalism (p. 2-16) but is an additional functional requirement that is simply imposed by the authors.
  • Sleep: the disconnection between blanket and internal states that is discussed should hold for dreamless sleep (unconscious) as well as dreaming (conscious). What then makes the phenomenal difference?

Minor:

  • 97: we have two consider
  • Invalid citation on l. 101, and l. 130, footnote 2 etc… many refs missing, also l. 988

Reviewer 3 Report

The paper suggests a physical approach based on information concepts to the study of sentience and mentality. It is claimed that the essence behind a Cartesian duality can be resolved within an approach that is based on a particular distinction between internal and external information states.
The concept is finally discussed in a philosophical context suggesting a particular ‘dual aspect monism’ completely grounded in the physical domain (‘Markovian Monism’).  Further the ideas offered  are compared to other information based approaches in the study of consciousness, as those advocated by Tononi et al (‘Integrated Information Theory’).  In particular it is argued that Tononi’s theory builds on an axiomatic structure (avoiding the ‘hard problem’), whereas the present concept is based on the senior authors previous suggestion of a free energy principle in  neural information processes which can resolve the axiomatic structure into a ‘physical explanation’.

However:

The paper is very hard to read and simply said, not really understandable.

  1. For example, it is hard to understand how ‘pre-theoretical notions can be articulated in terms of maths’ and how and why standard physical equations can just become reinterpreted in the view of a new ‘psychological-information’ representing variables, e.g. a thermodynamical potential representing ‘surprisal’ or ‘belief’ ?
    E.g. the Einstein mobility (formula 1.10, line 378), is this electrical mobility, or mass-particle mobility related by 1/m.gamma with gamma representing friction ? If so, what is the ‘friction’ of information, belief, surprisal’ etc…. ? (or its charge ?
    Line 378-384: It is granted that ‘nothing has been done but … assigning new units of measurements to standard equations ‘ ?? This reflects the overall strategy of the concept presented here. Physical relationships between fundamental physical concepts involve fundamental physical constants, what are the constants here ?
  2. The contents of the paper very much builds on what is called a ‘Markov blanket’. However, this term is not really explained or defined well at all. Figure 1 is not suitable for this, the
    insertion of implicit formulas is simply not necessary for it. The term comes from causal modelling in Network Dynamics and can be clearly defined. Is there any and if, what evidence for a Markov blanket in the information processing domain in the brain? This is in fact a question that can be expected to be answered by the authors experience in neural Network Dynamics. Why is not really contained in the present Assay?
  3. Line 298: formula 1.6, Fisher information metric: is simply inserted as many other relations, without any explanation or derivation, its distance metric claimed to be a ‘Kullback-Leibler divergence’, this at least needs to be explained (as many other notations), not shifted into the citation part.
  4. All mathematical or physical relations appear to be simply ‘inserted’ without any prior motivation or ‘building up’ principle, with only a very poor symbolic description, many indices, no explanation of these. Everything could be kept much easier . The glossary at the end does not help much there, it appears more like ‘a graphical repetition’, not an explanation at all. In summary, this style would definitely not pass any serious mathematical-physics journal.

    Figure legends that go beyond 30 lines are not suitable to attract inspection or information ‘at a glance’.

  5. Although the present work comes from a highly experienced group in physiology and neural modelling, the way the contents is represented is not convincing. The authors simply want to much at once, resolving Cartesian duality by a speculative interpretation of physical conceptions

  6. A major revision would be necessary to make this excursion more understandable and perhaps even plausible.

Round 2

Reviewer 1 Report

The authors have addressed all my comments and questions. I enjoyed the authors' arguments about split-brain. The authors' prediction, "action or behavior determines whether a split brain is one or two people", is interesting to empirically test as different theories would make different predictions on this matter. I look forward to see more discussions from this Markovian Monism view on consciousness. 

Reviewer 2 Report

I want to thank the authors for the updates to their manuscript and the comments to my questions.

Here some final concerns:

Footnotes:

I appreciate the footnotes and think they are a good idea. However, I believe that Entropy does not allow for footnotes. With that in mind the authors might want to consider an appendix with endnotes. I hope the footnotes will not just be re-inserted into the main text upon publication.

Nonequilibrium steady-state:

Thank you for comments to my questions. I still struggle to see how a human being over its life time (taking this to be the relative time scale for consciousness, as confirmed by the authors) may possibly be one persistent Markov blanket with a stable, unchanging attractor. More generally, I don’t think that human beings can be described as “one thing” over their entire lifetime. Philosophically, biologically, neurophysiologically, and physically one can argue that a human now is actually a different “thing” (system/set of elements) compared to a second ago. How does that square with one unchanging big dynamical attractor. If you have a way to briefly clarify this issue maybe you could add a footnote, or sentence or two.

Also, given that the equations do not hold outside the attractor on transients, how does that idea about the “time it takes for a particle of creature to return to its attracting manifold” (l. 932) fit in? Are systems conscious or unconscious when they are not on the attracting manifold? In what sense is there even a generative model when they are outside the attracting manifold?

Distinction between conscious and unconscious systems:

I see how the formalism could explain categorical differences between the phenomenology of simple organisms and complex organisms like human beings (l. 584-585).

I would still maintain that the formalism explicitly does not imply a “phase transition” between unconscious and conscious systems. The authors choose to impose complexity as a requirement. As said before, this requirement does not come from their formalism but is an additional (ad-hoc) postulate and I think this should be clarified more.

In any case, the claim that single-cell organisms are uncontroversially unconscious (l. 586) is simply not true, since obviously, e.g., panpsychists have a different view here, so at the very least it is controversial whether single-cell organism have phenomenology of any kind.

In addition to qualifying this statement, it would be worth making explicit what the authors deem necessary features of consciousness. Is consciousness = phenomenology, or do they presuppose functional properties? The authors mention intentionality and active inference.
I would argue that adding any additional requirement makes the account less conservative and parsimonious (except if “conservative” means pleasing the intuitions of most neuroscientists). The argument about combining cells into humans is not convincing to me as (1) the formalism actually does provide a way to resolve this issue even if we grant cells phenomenology due to the hierarchy of markov blankets and the functional properties that come with richer dynamics and (2) overlapping consciousnesses are in any case still a problem of the account (this would require something like IIT’s exclusion postulate to resolve).

Arguably, whether simple systems without intentionality and active inference etc. have some simple phenomenology, i.e. some notion of “what it feels like to be them” is not really relevant outside of metaphysical concerns (e.g. it wouldn’t matter much for ethics). Given that, however, I do not see why the authors seem to firmly stand on the “protomental“ side. Note here that I am not necessarily a "single cells have consciousness" advocate, I just think that taking the extrinsic information manifold as sufficient for some simple form of phenomenology (if such a connection is made at all) follows more directly from the authors' starting point than imposing additional distinctions between conscious and unconscious systems from outside of the formalism.

Reviewer 3 Report

Physics establishes relationships among its fundamental concepts. The physical meaning emerges by the functional dependence among the engaged expressions. I cannot agree with the authors that all of this is simply reducible to ‘a calculus of conditional probability distributions (e.g., Bayesian beliefs)’, eliminating essentially its physical meaning. This is a central, a core question, underlying the claims of this paper. I am sorry to say, that it does not help much to refer this to a table of glossary terms (as in the Arxiv paper). Physical constants arise when assigning units to relations of physical conceptions, not to ‘states’. The concept of ‘States’ of systems is part of a QM conception and more difficult to see in the present context.

As before, for example, taking equation 1.10 (line 345-350) again. There are forces, masses (involving mobility and viscosity, or/and charge transport), temperature and energy. Even if we assign some ‘temperature’ to a probabilistic network (in the nervous system), one would have to justify this and give an estimation of its expected size. Nothing like this is actually done. There is a conceptual ‘jump’ with relation to these concepts that is not really explained!

Equations 1.4 – 1.9 (chapter 4 and 5): The entire load of parameters and implicit relations are simply not necessary. The essence seems to be the non-linearity of information distance. Why do the authors not just give an example and demonstrate its significance for the present outline by some brain processes or explicit ‘information geometries’ ?

I do not see how one can ‘abstract away’ from physical meanings by simply replacing physical concepts by ‘mental properties’.

I do not think that the authors can or will change the style of the paper into a version allowing a much more obvious physical interpretation, although it is essentially based on physical arguments. However, a critical review is not a censorship. Particularly in view of the considerable experience of the authors and their previous contributions to the question of consciousness I would agree with its acceptance and leave the final decision up to the Editor in Chief.

In any case it is important that the now added footnote is clearly visible:

“The formal basis of the arguments in this – more philosophical – treatment of sentience and physics can be found in (Friston 2019). The current paper starts were Friston (ibid.) stops; namely, to examine the philosophical implications of Markov blankets and the ensuing Bayesian mechanics. For readers who are more technically minded, the derivations and explanations of the equations in this paper can be found in (Friston 2019) – using the same notation.’
